# Impaired Efferocytosis of Pericytes and Vascular Smooth Muscle Cells in Diabetic Retinopathy

**DOI:** 10.3390/cells14171349

**Published:** 2025-08-30

**Authors:** Tom A. Gardiner, Karis Little, Alan W. Stitt

**Affiliations:** 1School of Medicine, Dentistry & Biomedical Sciences, Queen’s University Belfast, Belfast BT9 7BL, UK; 2Centre for Experimental Medicine, Queen’s University Belfast, Belfast BT9 7BL, UK; k.little@qub.ac.uk (K.L.); a.stitt@qub.ac.uk (A.W.S.)

**Keywords:** diabetic retinopathy, efferocytosis, phagocytosis, vasculature, juxtavascular microglia, apoptosis, cell death, inflammation, neurovascular unit, autophagy

## Abstract

During diabetic retinopathy (DR), cell death has been characterized in all of the major retinal cell types, but was observed initially in the microvasculature, particularly the mural cells: pericytes and vascular smooth muscle cells (VSMCs). Indeed, our ability to identify the mural cell corpses called “ghost cells” within the vascular basement membranes (BMs) in eyes of diabetic patients and animal models is indicative that removal of dead cells, or efferocytosis (EF), is dysfunctional during this disease. EF is the process whereby apoptotic cells are eliminated through phagocytic engulfment and digestion and is essential to maintain tissue integrity and immune homeostasis. The process occurs in three distinct phases: finding and recognition, engulfment, and digestion, under the direction of “find me” and “eat me” signals and a large array of their cognate receptors and bridging molecules. Efferocytosis can be performed by many cell types, but most efficiently by professional phagocytes, and with such rapidity that the process is extremely difficult to detect in healthy tissues. As delayed EF is a recognized cause of autoimmune and inflammatory disease, mural cell death in DR may create inflammatory foci in the neurovascular unit (NVU). Here we discuss the basic mechanisms of EF in the context of DR and the impact of diabetic metainflammation on EF effector cell dysfunction.

## 1. Introduction

### Efferocytosis: An Essential Process in Tissue and Immune Homeostasis

Programmed cell death and its consequences are not only determined by the mode of death, but also by the timing and efficiency of corpse removal, a process now known as efferocytosis (EF) [1]. Indeed, it has been proposed that apoptosis, the principal mode of homeostatic cell death, is not complete until efferocytosis has been successfully accomplished [2].

The term EF encompasses all the phases in the detection, identification, and phagocytic removal of dead cells and is a crucial background function for physiological cell turnover, prevention of autoimmunity, and resolution of inflammation after injury or disease [3,4,5,6]. Caspase-dependent apoptosis represents the preferred mode of programmed cell death in homeostatic cell replacement and is characterized as “clean” and non-inflammatory in comparison to necrosis and other modes of cellular demise. However, the benign reputation of apoptosis relies on efficient purging of apoptotic bodies before progression to post-apoptotic or secondary necrosis can occur [2]. Therefore, efferocytosis constitutes the essential last step in apoptosis-based cell turnover, and similar mechanisms are recognized, even in primitive metazoan organisms [7]. Indeed, rapid and efficient efferocytosis carries such a high priority in normal tissues that many parenchymal cells can perform the function in an assumed role as non-professional phagocytes, although not as efficiently as when effected by dedicated phagocytes such as tissue macrophages [4,6]. The need for efficient EF is intimately tied to its dual actions in preservation of immune homeostasis and prevention of autoimmune and chronic inflammatory disease [8,9]: firstly to remove the threat of post-apoptotic necrosis, and secondly to initiate the anti-inflammatory signaling that is an inherent component of EF. Therefore, elucidation of the basic biology of EF and manipulation of its downstream immune mechanisms are currently under intensive investigation as potential avenues for therapeutic intervention in diseases such as systemic lupus erythematosus (SLE), multiple sclerosis (MS), and Alzheimer’s disease [10,11,12,13].

## 2. Efferocytosis in the CNS

In the CNS, including the retina, EF is chiefly performed by the microglia, the resident tissue macrophages and representatives of the innate immune response within the blood–brain barrier. Astrocytes also have a demonstratable capacity for phagocytosis, and their use of EF-related signaling and phagocytic engulfment in synaptic pruning is now well-recognized [14,15,16]. They share this function with microglia [17,18], employing some of the same receptors for the “eat me” signals displayed by redundant synapses and dead cells [15,19]. However, in EF of apoptotic neurons, astrocytes appear to be utilized chiefly for engulfment of relatively discrete fragments of apoptotic cells within their immediate locality, while microglia engulf the soma and larger segments of major cell processes [20]. In this regard, the intimate morphological association of astrocytes with synapses [21] and neural processes [22] affords them extensive access for clean-up of apoptotic remnants derived from the spatially expansive processes of large neurons, while the major workload of digestion and processing is handled by the microglia with their enormous capacity for phagocytosis and digestion [23]. In comparison, microglia, while lacking the extensive access of astrocytes to distant portions of neuronal cytoplasm, are freely mobile cells. As such, microglia may either extend their processes to reach EF targets, or when necessary, adopt an amoeboid morphology and engage in chemotaxis to home on the somata and other major portions of dead or dying neurons. Within the retina, Muller cells are the major macroglial cells and fulfil all the phagocytic functions that astrocytes perform in the brain and are likewise known to share their phagocytic duties with local microglia [24].

### Efferocytosis in CNS Development

Microglia have a similar, though not identical, phenotype and ontogeny with all tissue macrophages and a common embryonic origin in the yolk sac [25,26,27]. Importantly, like other tissue-specific macrophages, microglia maintain a self-sustaining population throughout adult life, with an insignificant contribution from circulating mononuclear cells [28,29]. Also, in common with other myeloid macrophages, microglial cells demonstrate a massive phagocytic capacity and an elaborate lysosomal apparatus (Figure 1A) that permits the phagocytic engulfment and intracellular degradation of several apoptotic cells simultaneously (Figure 1B). However, it has recently been shown that although digestion of enclosed dead cells may continue for an extended period, microglia may only engulf one additional apoptotic neuron at any given time [30]. During early development of the CNS, microglia are the only professional phagocytes within the neural ectoderm as they enter the primitive neural tube prior to neurogenesis, and are therefore on site during a period undergoing a high level of apoptosis in neural precursors [31,32]. However, it should be recognized that during the very early stages of neural development, the role of non-professional phagocytes is central, and in the primitive retina, the neuroepithelial cells function in this role [33].

Microglia infiltrate the neuroectoderm from local extensions of the primitive vasculature in an IL34/CSF1R dependent manner [32] and soon after the onset of neurogenesis [32]. Importantly, the inception of neurogenesis is closely followed by the commencement of programmed cell death, and Wu et al. cite IL34/CSF1R signaling and neuronal apoptosis as synergistic factors in microglial colonization of the CNS in zebrafish [32]. Therefore, it appears that “find me” signals released by apoptotic neurons may be partially responsible for the initial microglial colonization of the developing CNS. Although microglial ingress to the embryonic retina occurs a little later than when first noted elsewhere in the neural tube, they are known to enter both from the ciliary margin and from the primitive hyaloid close to the initiation of neurogenesis, an event associated with the commencement of apoptotic death in retinal ganglion cells [34,35]. The cell dynamics and chronology of EF has been extensively investigated in the developing retina and after injury although specific knowledge concerning the phagocytic receptors involved remains unclear [33]. However, as the same menu of EF-related receptors appears reasonably well conserved across different classes of macrophages, those employed by retinal microglia will presumably be similar, but selected according to the situation and the inflammatory environment [36].

## 3. Recognized Phases in EF

There are three definable phases of EF: finding and recognition of the cell corpse, phagocytic engulfment, and digestion. Each phase is precisely controlled by “find me” signaling molecules released by dying cells and the display of “eat me” molecules on their surface that may be recognized by several classes of cognate receptor expressed both by somatic cells and professional phagocytes (see summary in [Fig cells-14-01349-ch001]). Of course, the decision to engage an EF target cell requires the safety check of “don’t eat me” membrane markers, notably CD47, to prevent engulfment of healthy cells [37], a real possibility when professional phagocytes are deployed [38,39,40].

### 3.1. “Find Me” Signals

The “find me” signals that have been best characterized are the nucleotides ATP and UTP, the signaling lipids sphingosine-1 phosphate (S1P) and lysophosphatidylcholine (LPS), and the chemokine CX3CL1/fractalkine [5,41,42,43]. Importantly, the “find me” signals tend to be detected by a relatively small group of G-protein-coupled receptors, the diverse downstream signaling of which mediates profound changes in the gene expression profile of the phagocyte, preparing them for subsequent steps in the progression of EF [5,10,43]. Fractalkine is of particular importance in the CNS as it mediates a broad array of signaling responses between neurons and microglia [44]. In addition to its role as a “find me” chemokine [45], fractalkine has been shown to reduce microglial-mediated inflammation and exert neuroprotection in an animal model of DR [46].

### 3.2. “Eat Me” Molecules

In accord with the deprived energy status of dying/dead cells, “eat me” molecules are relatively few and represent normal cell components not displayed on the surface of healthy cells. Calreticulin, a calcium-binding protein with diverse roles in calcium homeostasis, cell adhesion, and as an endoplasmic reticulin chaperone also has immune roles in MHC display and as an “eat me” molecule as an activator of immunogenic forms of cell death. As such, calreticulin chiefly serves to attract antigen-presenting cells [47], and its externalization is not conducive to the homeostatic removal of apoptotic cells. In contrast, the principal “eat me” receptor in homeostatic efferocytosis is the negatively charged phospholipid phosphatidylserine (PS) [48,49], which can be bound by several different classes of EF receptors, some directly and others via soluble opsonins or bridging molecules [5,41,43,48]. PS is maintained on the cytoplasmic face of the plasma membrane (PM) through the action of ATP-dependent flippase enzymes that maintain phospholipid asymmetry in normal cells. However, during apoptosis, PS is flipped to the external leaflet of the PM by scrambalases that are ATP-independent, appropriate to the low-energy status of apoptotic cells [49,50].

### 3.3. Phagocytic Engulfment Receptors That Recognize PS

An elaborate array of engulfment receptors that bind to the “eat me” molecules are employed by different classes of phagocytes and varies according to the cell and tissue context, and the immune status of the immediate environment [5,6,41,43]. In microglia, the main groups of phagocytic receptors that bind PS directly can be grouped as follows: those with extracellular immunoglobulin domains, such as TIM family members—TIM1/4 [43,48,51,52] and the scavenger receptors CD300, stabilin-1, and RAGE (receptor for advanced glycation end products [11,53,54,55]). Likewise, the G-protein-coupled receptor BAI1 also binds PS directly and seems to be more important in microglia than for other classes of macrophage [48,52].

#### 3.3.1. TAM Receptors

Of those phagocytic receptors that utilize opsonins/bridging molecules, the TAM family of Tyro3, Axl, and Mer are receptor tyrosine kinases [56] and bind to PS-coated apoptotic cells via the bridging molecules Gas6 and Protein-S and have multiple downstream signaling targets [11,48]. Interestingly, galectin-3 (Gal-3) represents a less recognized third opsonin for MerTK-mediated phagocytosis that binds to glycoproteins on the surface of dead cells, rather than PS [57,58]. Gal-3 is of particular interest in the context of microglial phagocytosis [58], especially in retinal photoreceptor damage and DR [57,59,60,61], where its knockout is neuroprotective [59].

Axl and MerTK have overarching anti-inflammatory roles in the immune system, in which their dysfunction is associated with autoimmunity [62]. Within the immune system, Axl appears to predominate in dendritic cells, with MerTK as the principal TAM receptor expressed in macrophages, and Tyro3 in neither [63]. Also, studies in bone-marrow derived macrophages revealed that while both Axl and MerTK had overall anti-inflammatory effects, the expression of MerTK was increased in anti-inflammatory situations while Axl was reduced, a reciprocal relationship that held true for proinflammatory environments, where Axl increased and MerTK declined. Importantly, Axl expression increased in macrophages showing either M1 or M2 phenotypes, indicating that it responds to any form of inflammation or injury [63].

TAM receptors are also important in microglia, especially MerTK, as its dysfunction is associated with development of multiple sclerosis [64]. However, it remains to be seen whether the reciprocal expression of Axl and MerTK described in marrow-derived macrophages is reflected in microglia [63].

MerTK has emerged as the most important member of the TAM receptor trio in homeostatic EF, as it is the dominant form expressed in microglia and shows the most dysfunctional phenotypes in gene knockout animals, including outer retinal degeneration akin to RCS rats and photoreceptor degeneration similar to retinitis pigmentosa [43,65,66,67,68,69]. Similarly, polymorphisms in the MerTK gene have been strongly linked to MS and SLE [56]. Consistent with its status as an MS risk gene, studies on knockout mice show that MerTK is required for clearance of myelin debris and efficient remyelination following a demyelinating event [64].

In contrast to MerTK expression by microglia, AXL is upregulated in astrocytes after traumatic brain injury, where it promotes astrocyte switching to a phenotype that limits neuroinflammation and promotes efficient phagocytosis of dead cells and debris [70].

#### 3.3.2. Triggering Receptor Expressed on Myeloid Cells-2 (TREM2)

Discussion of phagocyte receptors that recognize phosphatidylserine requires mention of TREM2 as it has pervasive roles in microglial biology [71,72] and during neurodegenerative disease [73,74]. Importantly, dysfunctional genetic variants represent recognized risk factors for the development of Alzheimer’s disease (AD) [75,76,77], and identification of this disease connection proved instrumental in initiation of current research focused on this receptor [72]. In AD, dysfunction of TREM2 impairs the compaction and removal of amyloid plaques [78,79], consistent with its function as a receptor for amyloid beta [80]. In regard to EF, TREM2 is widely quoted as a PS receptor, although the two studies cited as support for the notion only identified the importance of TREM2 in phagocytosis of apoptotic cells and did not specifically study TREM2 binding to PS [81,82]. TREM2 does bind a broad class of damage-associated lipid molecules [83], however a 2015 study by Bailey et al showed that although TREM2 demonstrated some affinity for PS, it did not bind to the membranes of apoptotic cells [84]. These authors suggested that the activity of TREM2 in the clearance of apoptotic cells was due to its binding of apolipoprotein-E (ApoE) [85], which has been shown to associate with a broad range of lipid and membranous cell debris and facilitate their phagocytosis [86]. Therefore in the context of EF, TREM2 is obviously central to the promotion of multiple EF-related function, including chemotaxis [87], cell migration, and phagocytosis [71], but not as a specific PS receptor.

#### 3.3.3. Engulfment Receptors Involved in EF Target Cell Adhesion

An issue sometimes overlooked in reviews of this topic is that phagocytic engulfment in EF requires mechanical adhesion, and for dealing with large cargos, strong plasma membrane attachment to the actin cytoskeleton may be required [88]. Therefore, subtle differences in the nature of the initial attachment of PS-coated apoptotic cells by tethering receptors may require recruitment of coreceptors that provide essential downstream signaling and/or additional mechanical adhesion, depending on the size of the target and the nature of the opsonin [48,88,89,90,91,92,93].

In this regard, integrin receptors are of particular importance as they represent a class of phagocytic receptors with established roles in robust cell adhesion that can bind to PS on the surface of the apoptotic cell, but with firm anchorage of their cytoplasmic tails to the actin cytoskeleton [5,6,42]. Microglia are known to express αvβ3 and αvβ5 integrins and to utilize these in EF of apoptotic neurons via the bridging molecule MFG-E8 (milk-fat globule-EGF factor E8) [11,48] secreted by astrocytes [94]. Also, in regard to integrin-mediated binding to the cell corpse during EF, the importance of the complement system should be emphasized as the complement receptor CR3 is also known as the integrin receptor αMβ2 [95] (alternative names Mac-1 and CD11b-CD18 [95]) and plays a major role in both microglial-mediated synaptic pruning [18] and EF [6,10,42]. Release of the complement component C1q by neurons binds to PS that may be locally externalized on redundant synapses [96], initiating function-related pruning, or over the entire plasma membrane of cell corpses, precipitating deposition of C3 that is bound by CR3/αMβ2 [18,97] on microglia. In this regard, it should be noted that microglia themselves represent the main source of C1q in the CNS [98].

Interestingly, there is now some evidence that MerTK signaling may be important in integrin localization and the “inside-out” activation [99] typical of this class of adhesion molecules [100]. Also, in regard to its role in tethering apoptotic cells to the surface of the phagocyte, MerTK may have an advantage in bearing structural similarities to the cell adhesion molecule NCAM [93,101].

#### 3.3.4. Spoiled for Choice: Engulfment Receptor Diversity in EF

Although it is unclear why such a large spectrum of receptors are used in EF [6], it is assumed that this diversity can lend specificity to each situation and elicit appropriate downstream signaling. Some level of redundancy could also be expected with so many receptors targeted to PS; however, the TAM receptors and MerTK in particular appears indispensable [43,56] in the CNS. Also, the PS receptors BAI1 and Tim-4 have been shown to have distinctive and co-dependent roles in microglial phagocytosis of apoptotic neurons [52]. The situation is made more complex by the fact that not all receptors are expressed by all classes of phagocytes [43], although professional phagocytes appear to have a wider range of possibilities [6]. For instance, retinal pigment epithelial cells that serve as specialized [6], though not professional, phagocytes show absolute dependency on MerTK for the circadian phagocytosis of the photoreceptor outer segments [67]. Whatever the choice of EF receptors deployed, they must be able to distinguish dead from living cells, engage in membrane adhesion, and activate the signaling cascades that control the membrane dynamics and cytoskeletal changes required for phagocytic engulfment [4,5,6,41]. Receptor signaling following ligation of “eat me” molecules also increases the capacity for post-engulfment digestion, lysosomal processing of the products, and extrusion of those products that are surplus to the metabolic needs of the phagocyte, such as cholesterol derived from ingested cell membranes [6,42].

One important distinction that can be made in the classes of phagocytic receptors selected in any particular situation may be determined by whether the target has been opsonized by coating with immunoglobulins or complement in a proinflammatory environment. In this situation, macrophage Fc receptors are predominant and accompanied by other receptors only peripherally related to EF, and so not included in the present discussion [36,102].

## 4. Efferocytosis-Mediated Immune Modulation

Parallel to the internal signaling pathways responsible for the physical processes of cell disposal in EF are those that induce a pro-regenerative anti-inflammatory gene expression profile in the phagocyte, coupled with the release of anti-inflammatory cytokines, notably transforming growth factor beta (TGFβ) and interleukin-10 (IL10) [5,6,10,13,42,103]. Such anti-inflammatory signaling associated with EF is clearly its most important feature for health and of enormous consequence for disease causation if dysfunctional [4,5,6,8,9,12,13,42]. The anti-inflammatory actions of EF signaling in phagocytes commence early in the process, with the “find me” chemokine CX3CL1/fractalkine showing both anti-inflammatory and neuroprotective effects in microglia, even before engagement of engulfment receptor signaling [104,105]. Beyond the resolution of inflammation, the secretome of macrophages has been shown to actively promote wound healing and regeneration [13,106].

## 5. Impaired Efferocytosis in Diabetic Retinopathy

In the field of DR, we have been rightly preoccupied by identification of the pathogenetic mechanisms responsible for the dysfunction and death of retinal cells, with less attention given to what happens to the products of such death [107,108]. Cell death has been reported in both neural and vascular cells during DR [109], although the first notable diabetes-related cell loss was observed in capillary pericytes [110]. Many studies have reported increased cell death of retinal neurons in animal models of DR using various markers of apoptosis [111,112,113,114,115,116,117,118]. However, EF of the apoptotic neural cells has not been described, either by microglia or Muller cells, both of which can perform this function in retinal injury or disease [24,119]. Interestingly, pericyte loss was registered by the failure of EF in DR, resulting in persistence of uncleared corpses of dead pericytes described as “pericyte ghosts” within the capillary basement membranes in flat-mounted digest preparations of the whole retinal vasculature [110] (Figure 2A). Later studies on diabetic dog and human retina revealed the corpses of vascular smooth muscle cells (VSMCs) in the walls of retinal arterioles [120,121] (Figure 2B). As the corpses of each type of mural cell were most frequently observed in microscopic preparations as pockets of cell debris encased by discontinuous plasma membranes and basal lamina, they were initially considered as the remnants of cell death through primary necrosis [122]. However, our retrospective reappraisal of electron microscopic data from diabetic patients and animal models, presented in two previous reports [122,123], showed that retinal mural cell “ghosts” demonstrate ultrastructural features that are inconsistent with death by primary necrosis [122]. As expected, the pericyte and VSMC ghosts showed a spectrum of preservation states, but most contained abundant membranous profiles (Figure 3A,B), with the better-preserved examples offering convincing evidence of excessive autophagy [122] (Figure 3C,D). For a time, such autophagy was classified as a distinct mode of cell death [124,125]; however, as a constitutive survival mechanism employed by many cell types during starvation or other forms of stress [126], autophagy may constitute a final attempt to defer cell death before ATP depletion triggers either apoptosis or necrosis [122,127,128]. In fact, later recommendations by some of the investigators who initially proposed autophagy as a distinct mode of cell death caution against labelling any cell death situation as dependent on autophagy, outside experimental situations where strict mechanistic criteria can be met [129], and so the situation remains controversial [130]. It appears therefore that the status of cells showing excessive autophagy is ambiguous, with the situation made more uncertain by the mysterious fate of cell remnants in autophagic death, as the involvement of phagocytes appears to be excluded [124,126]. Interestingly, although the majority of the mural cell corpses had progressed to secondary necrosis [2,122], those detected during phagocytic engulfment appeared to have been engaged in excessive autophagy and cytoplasmic cleavage prior to phagocytosis [123]. In this regard, it should be noted that plasma membrane blebbing and cytoplasmic cleavage represent characteristic morphological changes in apoptosis [131]. Therefore, pericyte/VSMC death in vivo shows features of both autophagic and apoptotic cell death, suggesting that extreme autophagy precedes the initiation of apoptosis in DR-related mural cell loss [123]. This conclusion is consistent with the finding of apoptosis as the only mode of pericyte cell death demonstrated in human DR [132].

## 6. Juxtavascular Microglia (JVM) as Efferocytes of Dead Pericytes/VSMCs and Their Inhibition in DR

Our electron microscopic evidence identified the EF effector cells as juxtavascular microglia (JVM), a specialized population of microglial cells that were previously characterized in the brain, but not hitherto described in the retina [123] (see description below). It appears significant that although the majority of cell corpses identified as “ghost” cells within the vascular basement membranes in DR had progressed to post-apoptotic necrosis (Figure 3A,B), those observed while undergoing phagocytic engulfment by JVM (Figure 4) showed superior ultrastructural preservation and appeared to have been engaged at an earlier stage in the death process (Figure 5) [123]. As cells dying by apoptosis have been shown to release “find me” signals, such as ATP and UTP, very early in cell death [133], the presence of a JVM cell within the immediate or an adjacent neurovascular unit (NVU) may have overcome diabetes-related inhibitory factors in “find me” signaling or chemotaxis [134].

Specifically, the metainflammation induced by diabetes in the tissue environment of the CNS, including the retina [135,136], has been shown to downregulate expression of the P2Y12 receptor for purinergic “find me” signals (ATP and UTP) and impair JVM chemotaxis and the level of cell recruitment to sites of vascular injury [134,137]. Reduction of interferon gamma (IFγ) by anti-inflammatory treatment with dexamethasone restored P2Y12 levels and the normal vascular repair function of JVM [134,137]. Likewise, microglial responses were partially restored by insulin treatment and blood glucose levels approaching that of non-diabetic controls [134]. Therefore, JVM proximity and access could represent the deciding factors in whether successful EF can be performed on dying mural cells [123].

Also, in relation to inflammation-related inhibition of microglial responses in DR, a recent study by Pathak et al. has demonstrated additional recruitment of JVM in experimental DR that was significantly reduced in the absence of pentraxin-3 gene expression [138]. PTX3 is a pattern recognition molecule and modulator of innate immunity and inflammation, with multiple effects, including either enhancement or inhibition of phagocytosis, depending on the context [139]. Pathak et al. showed a marked increase in pentraxin-3 (PTX3) expression by the Muller glia during experimental DR [138]. Somewhat counterintuitively, this study also demonstrated that phagocytosis of apoptotic bodies was inhibited in human microglial cells treated with PTX3; the apoptotic bodies were derived from astrocytes and vascular endothelial cells induced with staurosporine [138]. Therefore, although PTX3 expression increased the number of JVM, it inhibited the phagocytosis of apoptotic bodies, consistent with the dual outcomes produced by this immunomodulator [140,141].

Despite of the importance of EF in limiting inflammation in diseases associated with cell death, studies of microglial EF engulfment receptors in DR are lacking. However, a recent study by Qi et al. reports MerTK impairment in experimental DR due to increased expression of ADAM17 [142]. ADAM17 is the principal matrix metalloproteinase for MerTK shedding that, in addition to decreasing the level of functional receptor on the efferocyte plasma membrane, further reduces its activity by liberating soluble MerTK (sMER) [143]. The soluble receptor has been shown to inhibit EF engulfment through binding and inactivation of the MerTK bridging ligand GAS6 [144].

### Juxtavascular Microglia in Normal Retina

Retinal JVM reside in close apposition to the blood vessels with their cell soma and nucleus immediately adjacent to the vascular wall and can either penetrate the glia limitans (GLs) of the astrocyte/Muller cell endfeet, or in some instances constitute an integral component of the GL [123] (Figure 6A). It is important to differentiate JVM from the perivascular macrophages that have been previously described and occupy an intravascular niche within the basement membranes of the retinal vasculature [145]. These cells were distinguished from microglia based on non-staining with the microglial immunomarker Iba-1 and electron microscopic confirmation of their exact location on the vascular side of the GL [145]. In comparison, the retinal JVM cells we observed in the phagocytic engulfment of dead or dying pericytes and VSMCs in DR were unambiguously located on the parenchymal side of the GL and also identified as a normal participant in the GL around normal retinal vessels [123]. Therefore, retinal JVM cells occupy precisely the same location as those previously described in the brain [146,147]. JVM in the brain have been investigated over a period of at least 30 years [137,146,147,148,149,150,151], and like other microglia, have a stable population showing little exchange with marrow-derived macrophages [148,151]. They are dynamic cells that can migrate along the microvessels to access sites of injury [149] and have been shown to physically seal sites of endothelial damage to restore the integrity of the blood-brain barrier [137]. However, it is now clear that although a certain population of microglial cells is always in contact with the vasculature, a recent study by Pathak et al. has shown that additional microglial cells may be recruited to the juxtavascular compartment in experimental DR [138]. Whether the role of JVM cells in microvascular repair described in the brain [137] is common to all microglia or attributable to a specific gene expression profile acquired by local signaling in the juxtavascular compartment is currently unknown. Joost et al. have proposed that JVM represent an integral component of the NVU in the brain [146], and we would extend that classification to the retina (Figure 6) [123] as they show cell junctions with adjacent astrocyte/Muller cell footplates and unique matrix attachments to the vascular basement membrane (Figure 6A).

## 7. The Challenge of Efferocyte Access to the Vascular Basement Membranes in DR

JVM also show the capacity to infiltrate the thickened vascular BMs [123] that characterize the retinal blood vessels in DR [108,120,152,153,154]. However, it is likely that such infiltration would be slower than in normal retinal vessels as JVM must overcome the inhibitory nature of matrix protein crosslinking and other modifications induced in the vascular BMs by advanced glycation end products (AGEs) during diabetes [152,155,156,157]. AGE-adducted basement membrane proteins are more resistant to the protease digestion that facilitates access to the matrix and can block the critical motifs necessary for cell attachment, migration, and survival [155,158,159,160]. Nevertheless, JVM cells appear to have a superior capacity for matrix infiltration than the local non-professional phagocytes that could theoretically fulfil the role of efferocytes for dead/dying mural cells.

## 8. Other Candidates for the Role of Efferocytes in EF of Apoptotic Pericytes and VSMCs

The other possible candidates for the EF of apoptotic mural cells are their partner endothelial cells within the vasculature as they exhibit significant phagocytic properties in situations of disease or injury [161,162]. Likewise, the Muller cells play major roles in EF during development and injury and have extensive access through their vascular endfeet at the glia limitans [24,33,163,164,165]. However, the previously mentioned increased expression of PTX3 or other proinflammatory changes induced in Muller cells by DR may alter EF-related genes [166,167,168]. Similarly, Muller cells are known to employ Gal-3 as a MerTK opsonin for phagocytosis in retinal degeneration, where it plays an anti-inflammatory role by reducing Muller cell activation [61]. However, in the context of DR, Gal-3 plays a pathological role in neuroinflammation and blood-retinal barrier dysfunction [59,169], a duality of outcome that may be due to the proinflammatory activation of Muller cells by diabetes-related metainflammation and AGEs [166,167,168]. The situation is further complicated by the presence of AGEs in DR as Gal-3 is also a well-characterized member of the AGE receptor complex [155,170], and so it is possible that the EF-related role of soluble Gal-3 as apoptotic cell opsonin for MerTK by Muller cells [61] is reduced by competition with AGEs in the diabetic retina.

Although microglial cells employ RAGE as a scavenger receptor for PS in phagocytosis of dead cells, there is no evidence that Muller cells do so in spite of the fact that RAGE expression in Muller cells is upregulated during DR [168]. RAGE signaling represents an ongoing source of vascular inflammation in diabetes [171], with its role made more complex through interaction with the increased production of its ligand S100B [168] and the secreted form of the nuclear chaperone high mobility group box-1 (HMGB1) [172] protein that has been implicated in inflammation and vascular injury in DR [173,174,175] (see the review by Steinle [173]). Importantly, HMBG1 has been shown to inhibit EF through binding of αvβ3 integrins and block their interaction with their opsonin MFG-E8 [176]. However, whether HMGB1-mediated inhibition of EF affects microglial cells during DR is currently unknown.

One last class of cells that represent credible candidates for the role of efferocytes for retinal vascular cells are the perivascular macrophages (PVMs) that lie within the BM envelope and on the endothelial aspect of the GL. Although our ultrastructural study provided no evidence of their engagement with mural cell corpses in DR, PVM are perfectly located and have the capacity to function as efficient efferocytes in the vasculature [145,177,178]. PVM have a scavenger function at the blood–retinal barrier and their precise location within the vascular basement membrane is well-described in a 2009 study by Mendes-Jorge et al. [145] (see comparison to JVM in our previous paper [123]). However, as PVM are also myeloid macrophages it may be presumed that they suffer the same diabetes-related impairments as microglia and peripheral macrophages in diabetic wound healing [179].

Therefore, in spite of the immediate proximity of such competent non-professional phagocytes, the majority of the dead pericytes in diabetic retinopathy pass into secondary necrosis with no evidence of attempted EF, except by JVM cells [122,123].

## 9. Proinflammatory Consequences of Failed EF in DR

Unfortunately, it is currently unknown whether EF is dysfunctional within the retinal neuropile. However, it can be speculated that EF of neuronal and glial cells also suffers some level of impairment, but without the additional matrix-related inhibitions imposed on EF of mural cells within the vascular BM.

As the retinal vascular cells represent a remarkably stable population with an extremely low level of turnover under physiological conditions [180], there is probably little need for homeostatic EF, with the need only apparent in situations of injury or disease. Although there is evidence for accelerated apoptosis in retinal vascular cells in DR [132], only the endothelial cells show evidence of regenerative capacity during this disease [180], and their increased turnover may not require local EF as apoptotic endothelial cells are probably removed by the circulation and scavenged in the liver. However, failed or dysfunctional EF of retinal pericytes and VSMCs has potentially deleterious immune consequences for the vessels harboring them in DR. Firstly, the release of necrotic cell products can incite local inflammation, with possible consequences, such as endothelial dysfunction and breakdown of the blood–retinal barrier. Secondly, the release of anti-inflammatory mediators by the efferocytes is negated, rendering sites of mural cell loss as proinflammatory foci within a host of retinal NVUs. Such inflammatory “hot spots” may increase local endothelial expression of adhesion molecules such as ICAM-1, thus creating potential sites for leukostasis-mediated endothelial injury [181,182]. Therefore, failure of EF in DR represents yet another function in which the vasculature suffers through diabetes-related dysfunction in other members of the NVU.

## 10. Further Insights from Published Data

Interrogation of data derived from publicly available datasets (retina—PMID: 33674409; brain—PMID: 37351595) from studies of the established DB/DB mouse model of type 2 diabetes showed interesting similarities and notable differences between retinal microglia and those from the neural cortex (Figure 7).

Significantly, the expression of TREM2, a gene with multiple roles in microglial biology, including chemotaxis, migration, and phagocytosis, was relatively reduced by diabetes in retinal compared to cortical microglia.

Several significant alterations were evident in genes associated with “find me” signaling. Expression of P2ry12, the gene that encodes the P2Y12 purinergic receptor, was more depressed by diabetes in retinal compared to cortical microglia, a change paralleled by reduced expression of a second purinergic receptor gene, P2ry6, in retinal microglia. As P2Y12 triggers microglial activation in response to the “find me” signals ATP and UTP in EF and P2Y6 increases phagocytosis in response to UTP [183,184], both changes could contribute to slowed and inefficient EF during DR. Similarly, reduced expression of the fractalkine receptor gene Cx3CR1 was evident in retinal microglia during diabetes compared to those from the cortex, a change that in addition to impairment of “find me” signal detection could have proinflammatory effects in the microglial cells [45,46].

A number of differences were noted in microglial expression of genes related to PS receptors or their opsonins. Cortical microglia showed a significantly lower expression of MerTK in the diabetic mice compared to controls, a change that was reflected in retinal microglia, but was statistically insignificant due to the relatively smaller population of microglial cells available in retinal tissue.

The gene coding for a second TAM receptor, the AXL kinase showed a low level of expression in the cortical microglia of diabetic animals but was almost absent in controls. However, retinal microglia showed a low but definite expression of AXL in both diabetic and control retina, but no measurable difference accountable to diabetes.

In contrast to TAM receptor expression in microglial cells, both MerTK and AXL showed a relatively high level of expression in the cortical astrocytes and Muller glia of the retina that was not significantly altered by diabetes, consistent with their role as non-professional phagocytes.

The scavenger receptor stabilin-1 that binds directly to PS was expressed by both cortical and retinal microglia, but its expression was relatively higher in those of the cortex. Stabilin-2 receptor showed no expression in microglia from either location.

The soluble opsonin/bridging molecules for PS tended to be increased in microglia. GAS6 was marginally increased in diabetic microglia, as was the integrin opsonin MFG-E8. However, the levels of MFG-E8 expression in Muller cells and astrocytes suggest that the Muller cells and astrocytes represent more significant sources of MFG-E8 than the microglia. MFG-E8 was highly expressed by Muller cells in control retina and increased during diabetes. In contrast, diabetes significantly reduced MFG-E8 expression in cortical astrocytes. Interestingly, treatment with exogenous MFG-E8 after traumatic or hemorrhagic brain injury is both anti-apoptotic and anti-inflammatory [185,186,187,188]. Although these studies did not seek to dissect the benefits of enhanced EF by MFG-E8, a recent study showing reduced inflammation and increased neuroprotection when microglial EF is potentiated by exogenous MFG-E8 suggests that the level of MFG-E8 may be limiting in situations with high levels of cell death [189]. The related issue of EF-associated microglial cell expression of the αvβ3 and αvβ5 integrin receptors that bind MFG-E8 is interesting. The genes encoding the beta subunits of the αvβ3 and αvβ5 (Itgb3 and Itgb5, respectively) are expressed by cortical microglia, although only Itgb5 was detected in those from the retina, and neither was altered in diabetic mice. However, Itgav, the gene coding for the alpha subunit designated CD51 and shared by αvβ3 and αvβ5 was undetected in microglia from either the brain or the retina (therefore not displayed in Figure 7). This apparent discrepancy can be explained by the observation that CD51 may only be induced in certain populations of macroglia or during inflammation [190]. Also, CD51 is expressed by both non-professional phagocytes (Leydig cells) and macrophages in the seminiferous tubules of the testis [191], a site of constant apoptosis and an associated need for EF [192,193]. Therefore, in the brain, it would be interesting to examine sites of adult neurogenesis for CD51-positive microglia, as these are locations with ongoing apoptosis with a continual need for EF [194]. Therefore, the expression of CD51 may represent a marker for microglia engaged in EF. Of course, if CD51 was only expressed in situations of cell death, it would probably require induction by the early” find me” signals to be available at the stage where phagocytosis commenced.

Expression of all three genes encoding the complement factor C1q component peptides C1qA-C were reduced by diabetes in retinal microglia compared to those from the cortex, although expression of the CR3 receptor genes (Itgam and Itgb2) was unaltered by diabetes in either the brain or retina. Remarkably, MEGF10, a scavenger receptor employed by astrocytes for synaptic pruning and efferocytosis [15,195], but not expressed by microglia [195,196], showed a relatively high level of expression in Muller cells compared to the cortical astrocytes. MEGF10 is a high-affinity receptor for C1q bound to PS on the surface of redundant synapsed or apoptotic cells, and although its expression was unaltered in diabetic mice, its use by Muller cells deserves further investigation.

In summary, the data described are consistent with impaired EF during DR in relation to both “find me” and “eat me” signaling and attributable to the metainflammation of the diabetic state and the influence of a proteome modified by non-enzymatic glycation. However, transcriptional alterations in particular genes occurring within the metabolome of the diabetic retina require cautious interpretation as they could result from factors unrelated to EF. Metabolic alterations or compensatory feedback due to dysfunction in the protein product caused by the addition of AGEs can have profound influences. It is also needful to examine the other roles played by proteins encoded by the altered genes. A pertinent example is the scavenger receptor stabilin-1 that shows increased expression in cortical microglia within our analysis. Stabilin-1 is a known receptor for AGEs and has important roles in the biosecretory pathway, endocytosis, and lysosomal function (see discussion by Twarda-Clapa et al. [197]), all of which are involved in the phagocytic functions engaged during EF. In short, the situation in regard to stabilin-1 embodies significant complexity that precludes any attempt to directly relate alterations in its transcription to EF in the diabetic retina. Therefore, there is an obvious need for more experimental data using models that permit dissection of the roles of the various players in EF and weighing of their relative contributions.

## 11. Conclusions

This perspective has reviewed current knowledge of EF in the CNS and highlighted impaired EF as a hitherto unrecognized component of the proinflammatory environment during DR. Specifically, we discuss evidence that EF of pericytes and VSMCs is delayed during DR and the proinflammatory consequences of such delay within the NVU. We also discuss JVM as the only identifiable efferocytes for apoptotic mural cells during DR and the unique challenges for EF when the target cells lie encased within a dense glycated extracellular matrix. The role of diabetes-associated metainflammation on EF-related functions of microglia is also discussed. Beyond the implications for vascular function in DR, impaired EF of dying mural cells emphasizes our lack of knowledge on the fate of the other cells of the NVU that are known to undergo apoptosis during DR, especially the nerve cells and Muller glia.

In general, the tendency for microglia in the diabetic retina is to assume a proinflammatory phenotype [198,199] akin to M1 macrophages that show impaired chemotaxis and phagocytosis of dead cells in diabetic wounds [179].

There is now significant evidence that hyperglycemia and the associated oxidative stress induce a persistent proinflammatory state in marrow-derived myeloid macrophages [200,201] that is refractory to restoration of euglycaemia and may be accountable to epigenetic changes manifested in metabolic memory [202] and “trained immunity” [203,204]. However, as the microglia are not derived from marrow-based progenitors, they may be less susceptible to such metainflammatory conditioning offering hope of successful restoration of functionality with appropriate treatments. Also, the period of exposure to chronic inflammatory stimuli required to induce such epigenetic change is unclear. This is especially true in diabetic patients for whom hyperglycemia is episodic, unlike the unremitting high-glucose environment of the animal models in which metabolic memory was characterized [202,205]. Interestingly, a recent study by Zhang et al. has shown that intraventricular injection of the opsonin MFG-E8 was able to promote microglial EF and exert neuroprotection after an acute ischemic insult [189]. This treatment also induced a generalized and lasting change in the microglial gene expression profile from a proinflammatory phenotype to one that was anti-inflammatory and neuroprotective [189]. The authors emphasized the importance of interferon regulatory factor-7 (IRF7) upregulation in the post-treatment microglial gene profile [206]. IRF7 is associated with the transition from a proinflammatory M1 macrophage phenotype to an anti-inflammatory regenerative M2 profile [206,207,208]. Indeed, at a fundamental level, in both development and adulthood, microglia can radically alter their phenotype in response to paracrine signals from their microenvironment [178,209], and as such present themselves as suitable targets for treatment, rather than as immutably committed to a particular fate. Indeed, little is known on how the microglial gene expression profile responds to changes in the metabolic environment of the retina as a result of improvements in patient care. This includes technical innovations in glucose monitoring and insulin delivery resulting in improved glycemic control for type-1 diabetes, and more so in view of the increased complexity of the therapeutic background in patients receiving treatment for type 2 diabetes [210]. In particular, agonists for the receptors of the incretin hormones [211], glucagon-like peptide-1 (GLP-1) and glucose-dependent insulinotropic polypeptide (GIP) have made a major impact and appear to hold promise beyond improvements in glycemic control, blood lipid profile, and cardiovascular outcomes [212]. The anti-inflammatory and neuroprotective actions of GLP-1 agonists in particular have shown benefit in models of neurodegenerative disease and promise in clinical trials [213,214,215,216]. In particular, the propensity of GLP-1 agonism to alter the activation state of microglia with a change towards an anti-inflammatory M2 polarization has been noted by a number of authors and has obvious relevance to the present discussion [214,215,217,218].

In spite of promising results with GLP-1R agonists in preclinical studies of DR [219], hopes for improvements in the treatment of the condition were initially tempered by early trials that showed worsening of retinopathy in a significant number of patients [212,220]. However, interpretation of the results from such large unfocused trials requires a nuanced approach, and readers are referred to an excellent assessment by Simo and Hernandez [219], who point out the broad inclusion criteria of the trials and factors such as preexisting retinal ischemia in some patients. They also cross-reference the worsening of DR status in trials of GLP-1R agonists to early results from the landmark Diabetes Control and Complications Trial, where rapid restoration of euglycaemia was associated with worsening of DR in patients with established retinopathy [221]. Ironically, GLP-1R-mediated improvements in cardiovascular function could potentiate the angiogenic stimulus and trigger proliferative DR in patients with preexisting retinal ischemia, as the overarching effect of diabetes is fundamentally anti-angiogenic (see discussion by Stitt et al. [222]). Simo and Hernandez also provide a helpful overview of preclinical studies of GLP-1 agonists in DR and emphasize the need for improved clinical assessment and patient selection to maximize benefit from these important agents [219]. An exciting aspect of GLP-1R agonists is that they exert positive effects on all the component cells of the NVU and may therefore have long-term benefits that are yet to be uncovered.

Future studies of EF in mural cells during DR will require the use of inducible pericyte cell death in a model that can be performed with and without a background of diabetes. This would permit the opportunity to study EF in a large number of cells occurring within a defined timeframe and permit genetic and therapeutic manipulation of the process with evaluation of its impact on inflammation within the NVU. Also, in spite of the distinctions between microglia and peripheral macrophages, there exists an ongoing opportunity to examine treatment-associated changes in microglial gene expression, mimicked using patient-derived macrophages implanted in autologous neural organoids derived from induced pluripotent stem cells.

In DR, as in many neurodegenerative conditions, the innate immune response has emerged as a causal component of many phenomena in which it was previously regarded as a simple reaction to the central pathology. Therefore, treatments that embody anti-inflammatory effects in parallel with their primary purpose such as GLP-1R agonists may have much to offer in prevention or amelioration of DR in type-2 diabetic patients. Epidemiological studies of DR in patient groups receiving general long-term anti-inflammatory medication show a lower incidence of the condition [223], and a 5-year evaluation of a non-steroidal anti-inflammatory drug, erroneously reported as an aldose reductase inhibitor, in diabetic dogs markedly reduced vascular basement membrane thickening, without altering parameters of advanced glycation, oxidative stress, or the polyol pathway [224]. Currently, it remains to be seen whether such broad anti-inflammatory strategies or novel approaches targeting specific EF-related gene products may prove useful in chronic diseases such as DR. Nevertheless, the phenomenon of ongoing cell death in all the cells of the retinal NVU during DR requires intervention to steer proinflammatory cell death events to pro-regenerative anti-inflammatory outcomes.

## Data Availability

No new data was created for this work.

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
