# Peer review of "Impaired Efferocytosis of Pericytes and Vascular Smooth Muscle Cells in Diabetic Retinopathy"

_cells, 2025, doi:10.3390/cells14171349_

Round 1
Reviewer 1 Report
Comments and Suggestions for Authors
Removing dead cells in DR prevents the accumulation of cell debris which might contribute to inflammation further contributing to retinal damages. The present review deals very well with mechanisms underlying efferocytosis including some discussion about the development of anti-inflammatory strategies or novel approaches targeting specific efferocytosis-related gene products. My only suggestion would be to include a schematic representation tha can summarize the main findings of the present review. Additionally, I would like to invite the authors to evidentiate the novelty of the present review in respect to additional papers in the literature. Further questions to be answered also need to be disclosed.
Author Response
Comments of Reviewer-1: Removing dead cells in DR prevents the accumulation of cell debris which might contribute to inflammation further contributing to retinal damages. The present review deals very well with mechanisms underlying efferocytosis including some discussion about the development of anti-inflammatory strategies or novel approaches targeting specific efferocytosis-related gene products. My only suggestion would be to include a schematic representation that can summarize the main findings of the present review. Additionally, I would like to invite the authors to evidentiate the novelty of the present review in respect to additional papers in the literature. Further questions to be answered also need to be disclosed.
Author response to Reviewer-1:
We thank the reviewer for these helpful suggestions and have highlighted the novelty of the work and what is lacking in our current knowledge in the first paragraph of the Conclusions section. We also include a schematic of the EF that should prove helpful to DR researchers who are unfamiliar with the dynamic stages of the process in the new Box-1. We also suggest future studies that should be considered to expand our knowledge of the topic and its relevance to future treatments of DR in the extended Conclusions section.

Reviewer 2 Report
Comments and Suggestions for Authors
This paper is sound and very well organised. I have only one minor request from the authors.
I would like to see a paragraph on the influence of efferocytosis zo neurovascular unit integrity. Which are the critical points? Is there a point of no return in this process? Also if microglia are converted from M1 to M2 phenotype succesfully, would you expect that the damage done by DR could be revertd? In yes, in what extent?
I am particularly interested in this issue ince I myself suffer from DR induced by Type 2 diabetes.
Author Response
Reviewer-2: Comments and Suggestions for Authors
This paper is sound and very well organised. I have only one minor request from the authors. I would like to see a paragraph on the influence of efferocytosis on neurovascular unit integrity. Which are the critical points? Is there a point of no return in this process? Also if microglia are converted from M1 to M2 phenotype successfully, would you expect that the damage done by DR could be reversed? In yes, in what extent? I am particularly interested in this issue since I myself suffer from DR induced by Type 2 diabetes.
Author response to Reviewer-2 comments
The comments of this reviewer were extremely insightful and pertinent, and we hope that our response is adequately expressed in the extended Conclusions. We have tried to address any suggestion of immutability in the state of microglia, especially in diabetic patients where the metabolic situation differs so radically from the steady-state of hyperglycemia in animal models. We have also added some discussion of the recent developments in therapeutic management of diabetes. Our collective view is that microglia are extremely responsive to treatment and possibly the most flexible cell type within the CNS. Suitable treatment should be able to push the inflammatory profile of microglia in diabetes toward the M2 pole of the spectrum. We hope our ideas for future work will also prove helpful.
